# Reprogramming metabolic pathways *in vivo* with CRISPR/Cas9 genome editing to treat hereditary tyrosinaemia

Francis P. Pankowicz[1,2,3], Mercedes Barzi[1,2], Xavier Legras[1,2], Leroy Hubert[4], Tian Mi[5], Julie A. Tomolonis[6], Milan Ravishankar[1,2], Qin Sun[4], Diane Yang[1,2,7], Malgorzata Borowiak[1,2,3,6,7,8,9,10], Pavel Sumazin[5,10], Sarah H. Elsea[4], Beatrice Bissig-Choisat[1,8] & Karl-Dimiter Bissig[1,2,3,6,8,9,10]

Many metabolic liver disorders are refractory to drug therapy and require orthotopic liver transplantation. Here we demonstrate a new strategy, which we call metabolic pathway reprogramming, to treat hereditary tyrosinaemia type I in mice; rather than edit the disease-causing gene, we delete a gene in a disease-associated pathway to render the phenotype benign. Using CRISPR/Cas9 *in vivo*, we convert hepatocytes from tyrosinaemia type I into the benign tyrosinaemia type III by deleting *Hpd* (hydroxyphenylpyruvate dioxigenase). Edited hepatocytes ($Fah^{-/-}/Hpd^{-/-}$) display a growth advantage over non-edited hepatocytes ($Fah^{-/-}/Hpd^{+/+}$) and, in some mice, almost completely replace them within 8 weeks. *Hpd* excision successfully reroutes tyrosine catabolism, leaving treated mice healthy and asymptomatic. Metabolic pathway reprogramming sidesteps potential difficulties associated with editing a critical disease-causing gene and can be explored as an option for treating other diseases.

[1] Center for Cell and Gene Therapy, Baylor College of Medicine, Houston, Texas 77030, USA. [2] Center for Stem Cells and Regenerative Medicine, Baylor College of Medicine, Houston, Texas 77030, USA. [3] Graduate Program, Department of Molecular and Cellular Biology, Baylor College of Medicine, Houston, Texas 77030, USA. [4] Department of Molecular and Human Genetics, Baylor College of Medicine, Houston, Texas 77030, USA. [5] Department of Pediatrics, Texas Children's Hospital, Houston, Texas, USA. [6] Graduate Program in Translational Biology and Molecular Medicine, Baylor College of Medicine, Houston, Texas 77030, USA. [7] McNair Medical Institute, Houston, Texas, USA. [8] Department of Molecular and Cellular Biology, Baylor College of Medicine, Houston, Texas, USA. [9] Program in Developmental Biology, Baylor College of Medicine, Houston, Texas 77030, USA. [10] Dan L. Duncan Cancer Center, Baylor College of Medicine, Houston, Texas 77030, USA. Correspondence and requests for materials should be addressed to K.-D.B. (email: bissig@bcm.edu).

The most successful treatments for monogenic liver diseases, diabetes and hyperlipidaemias rely on pharmacological blockade[1–3]. Pharmacological blockades can be quite effective for a period of time, but they also have numerous drawbacks, including incomplete inhibition of the target enzyme, compensatory upregulation of the target, off-target effects, patient non-compliance and drug interactions. In theory, gene replacement therapy or gene correction could be used to treat metabolic disorders[4], but it relies on genomic integration of the therapeutic gene for sustained expression. Integration can be facilitated by designer nucleases such as the bacterial type II clustered regularly interspaced short palindromic repeats/Cas9 (CRISPR-Cas9) system[5–7], which create double-stranded breaks (DSBs) in the DNA, but the nature of DSB repair poses a challenge: DSBs can be repaired by homologous recombination during the S/G2 phase of the cell cycle or, in post-mitotic cells, by the error-prone non-homologous end joining machinery[8–10]. Since most metabolic liver disorders present with a low mitotic index (<1%), the non-homologous end joining pathway is the one most likely to repair CRISPR-induced breaks, with the attendant risk of generating dominant-negative variants or novel epitopes in the process.

We decided to combine the power of CRISPR/Cas9 technology with the insight of pharmacotherapeutic approaches in a strategy referred to as metabolic pathway reprogramming: we use the CRISPR/Cas9 system to genetically delete or inactivate part of a disease-related pathway—a one-time treatment that results in permanent inhibition of the target enzyme. As proof of principle,

we applied metabolic pathway reprogramming to hereditary tyrosinaemia[11–13].

Hereditary tyrosinaemia type I (HT-I) is caused by a deficiency in fumarylacetoacetate hydrolase, which catalyses the final step of tyrosine catabolism. Fumarylacetoacetate hydrolase deficiency thus leads to an accumulation of tyrosine and toxic catabolites such as succinylacetone, resulting in a lethal form of tyrosinaemia (HT-I) (Fig. 1a). Since 1992, patients have been treated with nitisinone[2], which inhibits the second step of tyrosine catabolism, hydroxyphenylpyruvate dioxigenase (HPD). This pharmacological block is incomplete, so that although nitisinone treatment reduces the risk of HT-I patients developing hepatocellular carcinoma, the incidence of this cancer is still significantly greater in this population[14,15]. $Fah^{-/-}$ mice treated with nitisinone also suffer an increased risk of hepatocellular carcinoma, but this risk disappears when the mice are crossed with $Hpd^{-/-}$ (HT-III) mice[16]. We therefore hypothesized that a genetic deletion of $Hpd$ in the liver using CRISPR/Cas9 technology might be a more efficient therapy than an incomplete pharmacological block by nitisinone. This strategy of genetically blocking a gene other than the diseased gene as a treatment is the core of metabolic pathway reprogramming.

We here demonstrate successful metabolic pathway reprogramming by converting HT-I ($Fah^{-/-}$) into HT-III using an exon deletion strategy of the $Hpd$ gene in the liver. Edited ($Fah^{-/-}/Hpd^{-/-}$) hepatocytes display a growth advantage over non-edited ($Fah^{-/-}$) hepatocytes, and replace the entire liver in only a few weeks. Animals treated by metabolic pathway

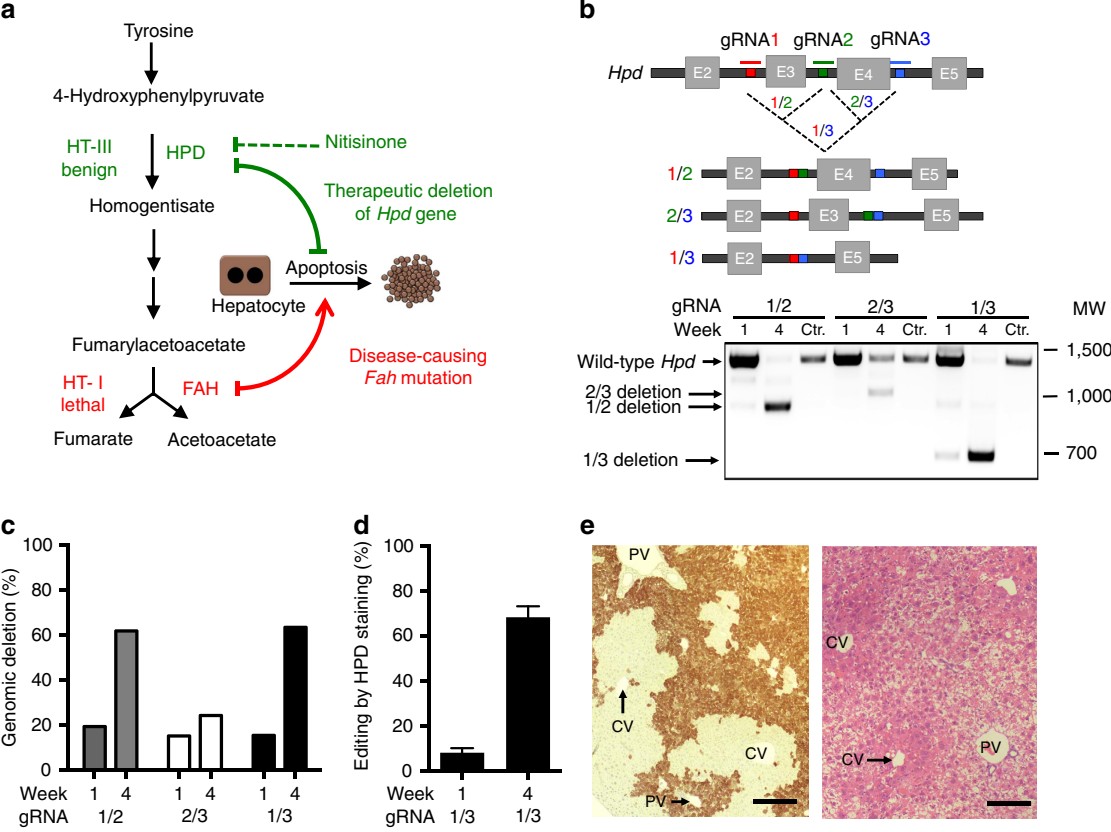

**Figure 1 | Metabolic reprogramming for HT-I.** (**a**) The therapeutic conversion of HT-I to HT-III can be accomplished genetically by deleting the $Hpd$ gene. (**b**) Schematic representation of the editing strategy using several gRNA pairs *in vivo* and validation by PCR of liver tissue. (**c**) *In vivo* editing efficiency determined by high throughput sequencing (pooled DNA libraries from three livers in each group). (**d**) Editing efficiencies measured by immunostaining for HPD using gRNA1/3 ($N = 3$). (**e**) Representative image of immunostaining for HPD (left) and haematoxylin and eosin stain (right) showing editing localized to the pericentral areas (CV). CV, central vein; MW, molecular weight marker given in base pairs; PV, portal vein. Scale bar, 50 μm. Bars represent mean (±s.d. for **d**).

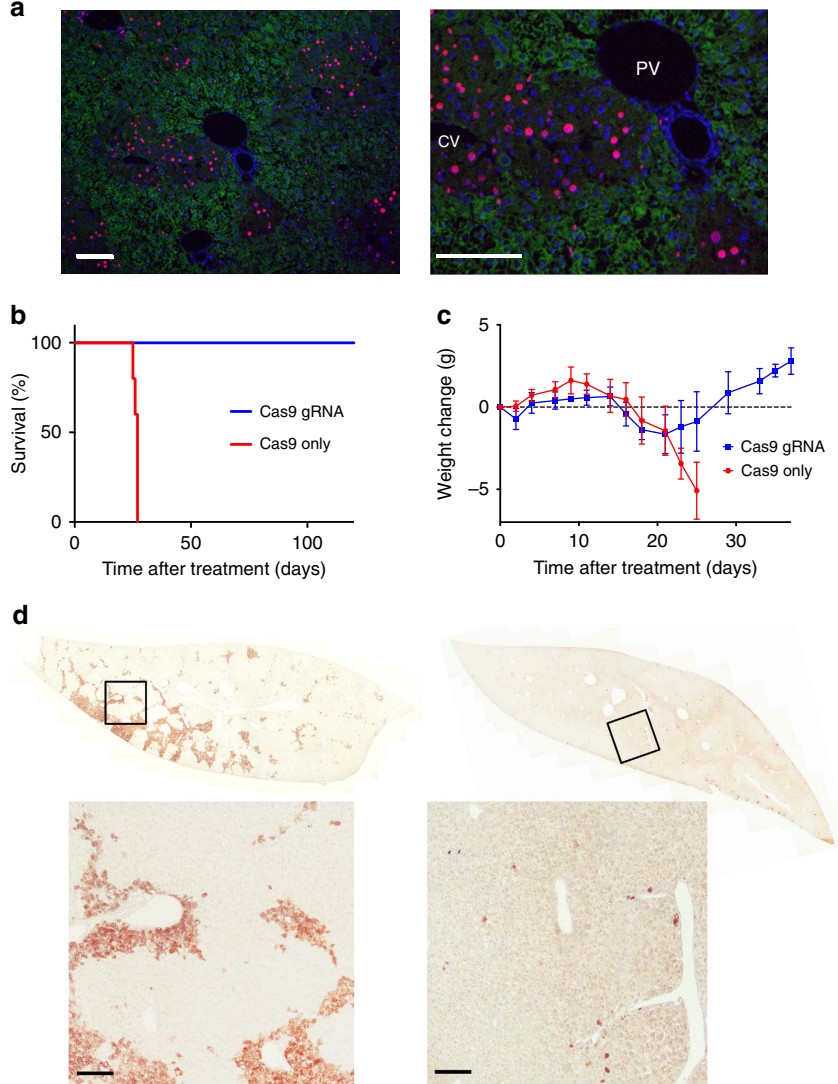

**Figure 2 | Proliferation of reprogrammed hepatocytes ($Fah^{-/-}/Hpd^{-/-}$) rescues HT-I lethality.** (**a**) Co-staining for EdU (red) and HPD (green), counterstain is 4,6-diamidino-2-phenylindole (blue). Proliferating, EdU-positive cells are HPD-negative. Kaplan–Meier survival curves (**b**) and body weights (**c**, mean ± s.d.) of tyrosinemic animals treated with either Cas9 only or both Cas9 and gRNA1/3 (each group $N=5$). Time after withdrawal from nitisinone (7 days after injection) is shown. (**d**) Range of repopulation by reprogrammed hepatocytes ($Fah^{-/-}/Hpd^{-/-}$) 8 weeks after injection. The mouse on the right has an almost complete repopulation by edited hepatocytes (>99%) and only a few non-edited ($Fah^{-/-}/Hpd^{+/+}$) hepatocytes are visible in the whole section. Black squares are amplified in the images below. Scale bar, 50 μm.

reprogramming are healthy, without pharmaceutical or dietary restrictions, and show an optimal metabolic profile compared with pharmacologically treated mice.

## Results

**Exon excision of the *Hpd* gene using CRISPR/Cas9.** To reprogram tyrosine catabolism, we designed short guide RNAs (gRNAs) to target the introns adjacent to exons 3 and 4 of the *Hpd* gene (Fig. 1b) using the online design tool (http://crispr.mit.edu). This allows critical exons to be excised without the risk of introducing potentially harmful mutations into the reading frame. We selected the 20 bp target sites based on location within introns (>100 bp from the exon) and predicted chances for off-target effects (Supplementary Fig. 1 and Supplementary Table 1), as determined by the software. To enhance *in silico* prediction of off-target effects, we also ran the software COSMID[17]. We evaluated exon excisions in NIH 3T3 cells transfected with

expression vectors coding a pair of gRNAs flanking the *Hpd* exons, as well as the Cas9 nuclease. All combinations demonstrated comparable deletions of expected sizes (Supplementary Fig. 2). Considering that genome-engineering techniques are dependent not only on sequence but also context (chromatin accessibility)[18–20], we further validated all three gRNA pairs *in vivo*.

We used the hydrodynamic tail vein injection method, which efficiently transfects up to 30% of hepatocytes in the murine liver[21], with 100–1,000-fold lower efficiency in the spleen, heart, kidney and lungs[22]. We injected $Fah^{-/-}$ mice with either Cas9 alone or with Cas9 and one of the three gRNA pairs. Animals were kept on nitisinone until injection and then weaned off the drug. Mice were killed and livers collected 1 or 4 weeks after injection to validate editing efficiency. The gRNA pair 1/3 generated the most efficient deletion as measured by PCR band shift (Fig. 1b); the wild-type *Hpd* band was barely visible by week 4, indicating almost complete editing. Deep sequencing

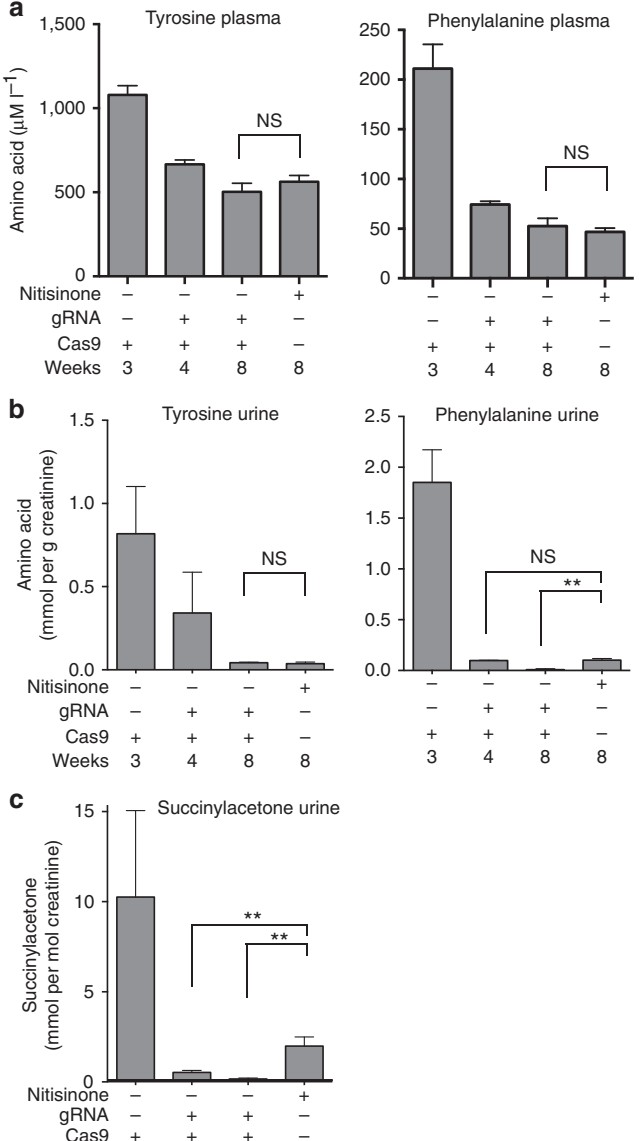

**Figure 3 | Metabolic profile of tyrosinemic mice treated with nitisinone or metabolic reprogramming.** (**a**) Tyrosine and phenylalanine levels in the plasma of mice treated with either nitisinone or CRISPR/Cas9, 4 and 8 weeks after therapy. Untreated mice were used as controls. Tyrosine, phenylalanine (**b**) and succinylacetone (**c**) in the urine compartment of the same mice as in **a**. *t*-tests were used to determine statistical significance. Bars represent mean ± s.e.m., group size (*N* = 3–5) mostly dependent on urine availability.

confirmed similar editing efficiencies (Fig. 1c), with only a small proportion of genetic inversions (Supplementary Table. 2). We also measured the off-target mutation rates of the predicted off-target sites for all three gRNAs used (Supplementary Table 1).

We decided to further analyse animals injected with Cas9 and the gRNA pair gRNA1 and gRNA3 (gRNA1/3). Western blotting of liver homogenate revealed a massive reduction of HPD protein in all treated animals when comparing 1 and 4 weeks after injection (Supplementary Fig. 3). The editing efficiencies at 1 and 4 weeks were 8% and 68%, respectively, when measured by immunostaining for HPD (Fig. 1d and Supplementary Fig. 4). Editing took place predominantly in the pericentral areas (zone III) of the hepatic acinus, where healthy hepatocytes could be found, in contrast to apoptotic periportal (zone I) hepatocytes

(Fig. 1e). This is consistent with pericentral hepatocytes being best transfected by hydrodynamic tail vein injections[21].

**Growth advantage and expansion of edited hepatocytes**. These results suggested that edited hepatocytes ($Fah^{-/-}/Hpd^{-/-}$) had a growth advantage over non-edited hepatocytes ($Fah^{-/-}/Hpd^{+/+}$). To prove expansion of edited hepatocytes, we injected the thymidine analogue EdU for 1 week before collecting CRISPR/Cas9-transfected livers. Double staining for EdU and HPD revealed that only edited hepatocytes ($Fah^{-/-}/Hpd^{-/-}$) expanded (Fig. 2a).

Since the expansion of edited hepatocytes might rescue the lethality of HT-I, we performed a survival experiment with a control (Cas9 only) and treatment group (Cas9 and gRNA1/3). Animals in the control group succumbed 25–27 days after withdrawal of nitisinone (Fig. 2b). The body weights, reflecting the dehydration that comes with HT-1, did not change significantly in either group until 21 days after nitisinone withdrawal, dropping only in the last few days before death (Fig. 2c). Mice in the treatment group were phenotypically indistinguishable from wild-type mice; they gained weight and did not require any nitisinone or dietary restrictions. We analysed mice 8 weeks after gRNA1/3 treatment and found an average editing rate of 92% by immunostaining. Strikingly, in some mice the diseased hepatocytes ($Fah^{-/-}/Hpd^{+/+}$) were almost completely (99%) replaced by healthy hepatocytes ($Fah^{-/-}/Hpd^{-/-}$) in just this 8-week period (Fig. 2d).

**Metabolic analysis of mice treated with CRISPR/ Cas9**. Finally, we analysed tyrosine catabolites and amino acids in the urine and plasma of genetically and pharmacologically treated animals. Mean tyrosine levels in the plasma decreased after editing and were on par with those of pharmacologically treated animals after 8 weeks, as were levels of the upstream amino acid phenylalanine (Fig. 3a). Urine amino acid analysis revealed a similar pattern (Fig. 3b), although in this compartment the excretion of phenylalanine was significantly lower 8 weeks after genome editing than after drug treatment. The toxic and pathognomonic catabolite succinylacetone was also significantly lower in genetically treated mice than in nitisinone-treated mice after 4 and 8 weeks (Fig. 3c).

**Discussion**

Here we provide proof of principle that metabolic pathway reprogramming is effective. By excising exons 3 and 4 of the *Hpd* gene, we successfully converted hepatocytes of tyrosinaemia type I into the benign type III. These hepatocytes displayed a growth advantage and repopulated most of the murine liver, thereby rescuing the lethal phenotype. Our metabolic analyses suggest that metabolic pathway reprogramming is superior to the gold standard drug treatment for tyrosinaemia type I.

If human hepatocytes display a similar repopulation rate, metabolic pathway reprogramming might be explored clinically for tyrosinaemia type I. For human patients, an alternative gene therapy approach would be preferable to the hydrodynamic transfection method used here. Successful use of CRISPR/ Cas9-mediated genome editing in murine models for human diseases[23–26] suggests the system holds promise for clinical applications.

Orthotopic liver transplantation is a therapy of last resort, but many patients with metabolic liver disorders are eventually forced to undergo it[27–29]. Unfortunately, there are twice as many patients on the waiting list as there are available organs[30]. Moreover, transplantation carries all the risks attendant on life-long immunosuppressive treatment[31]. Metabolic pathway

reprogramming holds several advantages over both gene replacement therapy and traditional pharmacotherapy. In contrast to gene therapy, there is no requirement for sustained expression of the wild-type protein of the disease-causing gene—a protein that is often recognized as foreign and triggers an immune response, thereby limiting long-term expression. Although the bacterial Cas9 is also likely to cause an immune response[32], short-term immune suppression (a few hours/days) should suffice for a definitive, lifelong genome editing.

This approach is likely to be more efficient than strategies relying on homologous recombination in the quiescent liver due to the organ's low mitotic index[33], and genetic manipulation produces a more complete blockade than pharmacological approaches. An additional benefit is that off-target effects would be limited to the edited organ, in this case the liver, rather than producing systemic side effects observed with drugs.

More importantly, metabolic pathway reprogramming should be applicable to enzymes that are currently not targetable by small molecules; the strategy is highly adaptable, and with the proper knowledge could be applied to metabolic problems in the liver involving, for instance, triglyceride metabolism, glucose homeostasis, cholesterol synthesis or the urea cycle. The benefit to patients with life-threatening metabolic disorders, such as homozygous familial hypercholesterolaemia or lysosomal storage disorders, could be enormous. The advantages of a permanent one-shot treatment could outweigh the potential risks of genome editing, offering new therapeutic vistas for many patients.

## Methods

**Animal experiments.** All animal experiments were approved by the Baylor College of Medicine Institutional Animal Care and Use Committee. The tyrosinaemic FRG ($Fah^{-/-}$ $Rag2^{-/-}$ $Il2rg^{-/-}$) strain[34,35] was kept on 16 mg l$^{-1}$ nitisinone in the drinking water before experiments. A unit of 20 μg of DNA constructs were introduced in 6- to 8-week-old mice of both genders by hydrodynamic tail vein injection, using 10% w/v saline over 5–8 s. Following the injection, mice were weaned off nitisinone over 7 days, and where indicated weighed every second day thereafter. The thymidine analogue EdU was supplied by the drinking water (1 mg ml$^{-1}$) and injected daily (intraperitoneal 1 mg per mouse) for 7 days before euthanization in some mice.

**Construction of DNA vectors.** gRNA sequences were selected using the online tool (http://crispr.mit.edu). Sites were designed to cut in introns at least 100 bp away from critical exons. The selected gRNA-targeting sequences are shown in Supplementary Fig. 1. Annealed oligonucleotides (Sigma) were ligated into the pX330 vector using standard molecular cloning techniques. pX330-U6-Chimeric_BB-CBh-hSpCas9 was a gift from Feng Zhang, Addgene plasmid # 42230 (ref. 36).

**Cell culture and PCR.** NIH/3T3 cells (a gift from Margaret Goodell) were cultured in Dulbeccos modified Eagles medium supplemented with 10% fetal bovine serum, 100 U ml$^{-1}$ of penicillin, 100 μg ml$^{-1}$ of streptomycin and 250 ng ml$^{-1}$ of Fungizone antimycotic (Invitrogen). Media was replaced every 2 days and cells were subcultured twice per week using 0.25% (w/v) Trypsin–0.53 mM EDTA solution. In all, $5 \times 10^5$ cells were nucleofected with 300 ng total DNA in P3 Primary Cell Solution (Lonza) using the 4D Nucleofector program EN-158. Cells were collected after 48 h and genomic DNA was extracted using the DNeasy Blood and Tissue Kit (Qiagen). PCR amplification was performed using Phusion polymerase (NEB, cat# M0530L). Primer sequences are given in Supplementary Fig. 1.

**Liver histology and immunohistochemistry.** Mice were killed and 20–200 mg fresh liver samples were immediately snap frozen in tubes. Remaining samples were fixed with 4% paraformaldehyde overnight for paraffin blocks. Paraffin sections were used either for haematoxylin and eosin or immunostaining.

Sections were treated with Target Antigen Retrieval Solution (Dako). Endogenous peroxidase was blocked using 3% hydrogen peroxide followed by an overnight incubation with primary mouse anti-HPD antibody (Santa Cruz sc-271672) diluted 1:100. Biotinylated secondary antibody was incubated for 30 min. Detection of HPD was performed using the M.O.M Vectastain kit and DAB detection kit (Vector Laboratories) according to the manufacturer's recommendations. Haematoxylin was used for counterstaining.

For immunofluorescence detection, endogenous biotin was blocked using Avidin/Biotin blocking kit (Vector laboratories) before incubation with biotinylated secondary antibody. Streptavidin-AlexaFluor488 (Thermo Fisher Scientific, cat# S32354) was incubated for 30 min at a dilution of 1:1,000. EdU was detected using the Click-iT Plus EdU Alexa Fluor 594 Imaging Kit (Molecular Probes) according to the manufacturer's instructions.

**Quantification of editing efficiency and expansion.** Images of immunostaining for HPD were taken using a Zeiss Axioplan 2 microscope with Photometrics CoolSnapHQ camera. A total of 10 random viewing fields per mouse were used to count the number of HPD-negative cells and total number of cells using the ImageJ software (W. Rasband, National Institutes of Health, Bethesda, MD; http://rsb.info.nih.gov/ij). To determine the proportion of cells that expanded at weeks 4 and 8, composite images of three liver sections per mouse were generated using Adobe Photoshop. The percentage of HPD-negative cells per liver section was determined through thresholding and area measurement (minimum pixel number: 10) with ImageJ.

**Western blotting.** Western blotting was performed as described previously[4]. Briefly, tissue from three mice per group was homogenized in RIPA buffer (Sigma, catalogue# R0278-50 ml) containing proteases inhibitors (Roche, catalogue# 04693159001). A unit of 30 μg of total protein was electrophoresed in a NuPAGE 4–12% Bis Tris Gel (Invitrogen, catalogue# NP0336BOX) and transferred to a polyvinylidene difluoridemembrane (Millipore, catalogue# IPVH00010). The blot was then blocked in 3% BSA, followed by primary antibody exposure. Anti-HPD antibody (Sigma, catalogue# HPA038322-100UL) was diluted 1:500, and donkey anti-rabbit IgG/horseradish peroxidase (Jackson Immunoresearch Labs, catalogue# 711-165-152) was used at 1:30,000. The membrane was imaged using Amersham ECL Western Blotting Detection Reagent (General Electric Healthcare Life Sciences, catalogue# RPN2106).

**Metabolic analysis.** Blood was collected using retro-orbital puncture of three mice for each group. Urine was collected over 24 h using metabolic cages (Techniplast, catalogue#3700M022). Urine succinylacetone was analysed using standard methodologies as described previously[37]. Briefly, urine was mixed with an internal standard (13C succinylacetone, Cambridge Isotope, MA), and oximation was performed with hydroxylamine hydrochloride at pH > 12, before adjusting pH to 1. The mixture was loaded on to a Chem Elut cartridge (Agilent, catalogue#12198006), eluted with ethylacetate and diethyl ether, dried under nitrogen and derivatized with Regisil. Trimethylsilyl derivatized compounds were separated on an Agilent 6890 Gas Chromatograph with a capillary GC column. Detection was performed by electron impact mass spectrometry (5973 MSD, Agilent, CA).

Plasma and urine amino acid analyses were performed on a Biochrom 30 HPLC amino acid analyser per standard protocols[38]. Physiological amino acid standards were used to determine analyte concentration and analysed using the EZchrom Elite software.

**Deep sequencing.** Genomic DNA was extracted from three different frozen liver sections per mouse. DNA libraries were prepared from ~1 ng purified PCR products using the Nextera XT Library Prep Kit (Illumina, catalogue# 15032350). A total of 10 candidate off targets were identified using CRISPR design tool[39] for each gRNA pair, see Supplementary Table 1. These regions, in addition to *Hpd*, were amplified by PCR; see primer identities in Supplementary Table 3. Resulting cDNA libraries were sequenced on Illumina NextSeq500 at >10,000 × coverage at amplified regions. Sequenced fragments were aligned to the reference (mm10) using BWA-MEM[40], identifying mutations, deletions, insertions and inversions at the amplified regions. All identified gaps and inversions were used to implement alternative reference regions, which were then used for fragment realignment together with reference regions. Reads aligned to the reference and to engineered gRNA-altered genomes were collected and quantified to identify the proportion of cells with deletions and inversions at *Hpd* and at candidate off-target genomic regions; see Supplementary Information for their quantification.

**Statistics.** Sample sizes for experiments were determined by estimated differences between groups and, in the case of survival rates, ethical considerations. No randomization of animals before allocation to experimental groups nor blinding of experimental groups was done. Statistical analysis was performed using PRISM version 6.0 software (Graph Pad software) using $t$-test, Mann–Whitney test or analysis of variance. Statistical significance was assumed with a $P$ value < 0.05 (*), $P < 0.01$ (**) and $P < 0.001$ (***). Bars in graphs represent mean ± s.e.m. unless noted otherwise. Group size ($n$) represents biological sample size.

**Data availability.** Sequencing data supporting the findings of this study have been deposited in the European Nucleotide Archive under ENA accession code PRJEB14753. The authors declare that the remaining data supporting the findings of this study are available within the article and in the Supplementary Information files.

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

## Acknowledgements

We thank V.L. Brandt and C. Gillespie for critical comments on the manuscript and Baylor Genetics for support of metabolic studies. K.-D.B. is supported by the National Heart Lung and Blood Institute (NHLBI) grant R01HL134510, the Texas Hepatocellular Carcinoma Consortium (THCCC) (CPRIT #RP150587) and the Diana Helis Henry and Adrienne Helis Malvin Medical Research Foundations. M.B. is supported by the McNair Medical Institute Foundation. F.P.P. and B.B.C. were supported by T32HL092332, The DLDCC is supported by P30CA125123. CMM core facility of Texas Medical Center Digestive Disease Center (P30-DK56338).

## Author contributions

K.-D.B. and F.P.P. designed the experiments; F.P.P. and B.B.-C. performed *in vivo* experiments; M.M.B. did immunostaining; J.A.T., D.Y., and M.B. prepared libraries and run sequencing; X.L. did western blotting and PCR; S.H.E., Q.S and L.H. performed analysis of catabolites and amino acids; T.M. and P.S. did bioinformatics. All authors read and approved the final manuscript.

## Additional information

**Competing financial interests:** The authors declare no competing financial interests.

