## [Peer Review File · Nature Communications]

Reviewers' comments:

Reviewer #1 (Remarks to the Author):

I have previously reviewed this manuscript by Pankowicz, et al. describing the use of CRISPR/Cas9 technology to treat hereditary tyrosinemia in a mouse model. Rather than reverse the causative mutation itself, the authors elected to mimic the current standard of care provided by treatment with the medication nitisinone. This drug inhibits an upstream step in the catabolic pathway of tyrosine and eliminates the secondary accumulation of succinylacetone in patients, mimicking another form of hereditary tyrosinemia caused by hydroxyphenylpyruvate dioxygenase deficiency (HPD). Using CRISPR/Cas9 technology to delete the gene for this enzyme via direct injection of appropriate gRNA pairs. Somewhat surprisingly, the entire liver of the treated animals eventually was repopulated by hepatocytes deleted of the HPD gene, suggesting a replicative advantage for the new cell type. Not surprisingly, the biochemical phenotype was shifted to an isolated tyrosinemia (type III), consistent with the human disease caused by the same gene defect. These mice had normal survival short term survival in a time frame in which the parent animals died without therapy. Biochemical correction was superior to that obtained with nitisinone treatment.

In the original version I identified a few weaknesses including documenting off site integration of editing vectors and gene inversions as well as deletions. The authors have added additional data that address these concerns. They have also added additional information on longer term outcome in the treated animals. This makes the manuscript much stronger and likely to be of broad interest to the metabolic and probably broader scientific community. Finally, they have included additional discussion on possible expansion of this therapeutic approach to other diseases. While the discussion is in general appropriate, one example in particular is not particularly relevant. The authors refer to substrate reduction therapy in urea cycle defects and particular carbamylphosphate synthetase deficiency. Substrate reduction will not be effective in this setting. I suggest that they instead offer examples of the lysosomal storage diseases where drugs focused on substrate reduction are already in use or clinical trials (ie, Gaucher and Fabry diseases).